# Deep Learning-Based Transmitter Localization in Sparse Wireless Sensor Networks

**DOI:** 10.3390/s24165335

**Published:** 2024-08-18

**Authors:** Runjie Liu, Qionggui Zhang, Yuankang Zhang, Rui Zhang, Tao Meng

**Affiliations:** 1National Supercomputing Center in Zhengzhou, Zhengzhou 450001, China; ierjliu@zzu.edu.cn (R.L.); yuankang_z@gs.zzu.edu.cn (Y.Z.); 2School of Computing and Artificial Intelligence, Zhengzhou University, Zhengzhou 450001, China; 3Zhengzhou Radio Center, Henan Provincial Department of Industry and Information Technology, Zhengzhou 450051, China; zhangr@163.com (R.Z.); 18538178867@163.com (T.M.)

**Keywords:** transmitter localization, deep learning, wireless sensor networks, location awareness

## Abstract

In the field of wireless communication, transmitter localization technology is crucial for achieving accurate source tracking. However, the extant methodologies for localization face numerous challenges in wireless sensor networks (WSNs), particularly due to the constraints posed by the sparse distribution of sensors across large areas. We present DSLoc, a deep learning-based approach for transmitter localization in sparse WSNs. Our method is based on an improved high-resolution network model in neural networks. To address localization in sparse wireless sensor networks, we design efficient feature enhancement modules, and propose to locate transmitter locations in the heatmap using an image centroid-based method. Experiments conducted on WSNs with a 0.01% deployment density demonstrate that, compared to existing deep learning models, our method significantly reduces the transmitter miss rate and improves the localization accuracy by more than double. The results indicate that the proposed method offers more accurate and robust performance in sparse WSN environments.

## 1. Introduction

With the rapid development of wireless communication technology, transmitter localization technology plays a crucial role in various fields, including military reconnaissance, emergency rescue [1], and radio interference monitoring [2]. The core goal of transmitter localization technology is to accurately determine the location of the signal source, thereby enabling effective tracking and management of the signal source, which is essential for ensuring the security of the electromagnetic environment. Due to its low cost and strong adaptability [3], wireless sensor networks [4] are widely used. By deploying WSNs, abnormal transmitters can be captured efficiently and quickly over a large area, achieving spectrum monitoring [5,6].

Traditional transmitter localization methods [7,8,9,10], such as time of arrival (TOA), time difference of arrival (TDOA), and angle of arrival (AOA), require complex hardware support and are difficult to deploy on a large scale. Distance measurement methods based on received signal strength (RSS) are affected by non-line-of-sight (NLOS) propagation [11]. Traditional RSS-based localization methods rely on RSS measurements and predefined wireless propagation models [12,13]. However, the complexity of real-world environments is challenging to quantify using propagation models, especially in large urban areas where buildings significantly impact signal propagation. This type of environment makes it challenging to correlate the estimated transmitter location with the actual position [14]. Other approaches, such as those utilizing routing protocols for optimization [15] and nonlinear system methods [16,17], can also be effective. However, these methods may involve challenges related to underlying hardware design or parameter estimation errors.

In recent years, advancements in deep learning technology have offered new solutions for the problem of transmitter localization in the field of wireless communication. By combining advanced neural network models and innovative data-driven methods to learn the geographic characteristics and propagation features of the region of interest (ROI), the fingerprint mapping relationship between data and location can be perceived, thereby achieving more accurate localization results. Several studies have shown the advantages of this learning-based localization technology. Firstly, the study in [18] utilized RSS data from different locations for deep learning training. The model employed a pre-trained multi-layer perceptron to extract features and introduced a hidden Markov model-based fine localization method to directly predict the transmitter’s location. The test results showed effectiveness in small-scale indoor environments but poor performance in outdoor environments. Recently, applying neural network approaches from image processing for transmitter localization has become a trend. DeepTxFinder [19] divided the ROI into grids, converting the RSS from sensors into corresponding pixel values for each location, and used convolutional neural network (CNN) for training. The fully connected layer was then used to predict the number and coordinates of the transmitters. This method was effective in densely populated small-scale outdoor areas. DeepMTL [20] improved upon DeepTxFinder by addressing the training difficulties and poor generalization caused by the fully connected layer. It transformed the transmitter prediction stage into an image processing task, where multiple transmitters correspond to local heatmaps in a 2D array. An improved YOLO structure was used to identify transmitter locations, enhancing localization accuracy and generalization performance. However, it still required a high-density sensor deployment, with a training density of 2%. TL;DL [21] utilized a U-net structure with a large receptive field to transform the sensor RSS matrix into a transmitter location matrix, using the argmax method to find transmitter locations. They also proposed a data augmentation method for handling missing sensor data. Experiments demonstrated the effectiveness of this approach with relatively sparse sensor deployments, but also showed insufficient localization accuracy over large areas and a higher transmitter miss rate.

To address the shortcomings of existing research in transmitter localization for spectrum monitoring, we aim to achieve high-precision localization with fewer sensors deployed over large areas, while reducing the transmitter miss rate. Based on the synthesis of prior literature, we consider that an area larger than one square kilometer can be defined as large areas, and WSNs with a grid deployment density less than 0.1% can be defined as sparse. Existing grid-based methods for large-scale areas often lead to significant localization errors, particularly when WSN density is low.

We propose DSLoc, a convolutional neural network model specifically designed to work with sparse sensor data. By incorporating efficient feature enhancement modules and an improved high-resolution network (HRNet) [22], DSLoc provides a more precise transmitter localization method. Our approach uniquely addresses the research gap by using a finer-grained grid division, even in extreme cases of low-density sensor deployment. For instance, we divide a 2 km × 2 km area into a 384 × 384 grid with only 0.01% sensor deployment density. Another key innovation of DSLoc is the use of an image centroid-based localization method. This approach determines the precise location of transmitters by calculating the centroids of heatmaps, significantly enhancing localization accuracy. Compared to state-of-the-art (SOTA) localization models, DSLoc achieves more than double the improvement in localization accuracy while improving overall performance. The proposed method’s effectiveness is validated through rigorous ablation and extension experiments. By addressing the critical challenges of sparse WSNs, DSLoc provides a robust and high-precision solution for transmitter localization over large-scale areas.

## 2. Problem Formulation

The transmitter localization process described in this paper is illustrated in Figure 1. In a region of interest, several sensors are sparsely deployed for spectrum monitoring. The RSS measured by the wireless sensor network is used to detect the transmitter’s location. The positions of the sensors and the corresponding normalized RSS measurements are mapped to a single-channel H×H two-dimensional matrix, or H×W matrix depending on the actual situation, where H is the height of the matrix and W is the width. The pixel values represent the normalized RSS, with grid cells lacking sensors assigned a pixel value of 0. The transmitter data are similarly mapped using this grid-based approach. However, unlike the sensor matrix, assuming the transmitter is located at $\left(x_{i},{\;y}_{i}\right)$, a two-dimensional Gaussian distribution is used to expand the transmitter matrix into a single-channel 2D probability matrix heatmap [23] around the true location, as shown in the following equation:(1)$$P\left(x,y\right)=A*exp\left\{-\frac{1}{2\sigma^{2}}\left[{\left(x-x_{i}\right)}^{2}+{\left(y-y_{i}\right)}^{2}\right]\right\},$$
where *A* is a configurable constant.

The proposed model, DSLoc, learns to map from the sensor pixel matrix to the predicted transmitter heatmap. The ROI is fixed, and the grid size H × H is adjusted as needed. For instance, mapping a 2 km × 2 km area to a 384 × 384 grid results in an actual localization error of 2 km/384 = 5.2 m if the network’s error is within one grid cell. As shown in the following equation:(2)MLE=Mean(p∗d),
where MLE represents the mean localization error, d represents the Euclidean distance between the target point and the predicted point, and p represents the mapping of each pixel to the actual scale. A transmitter that is present but not predicted in the heatmap is defined as a missed detection. Miss rate is the ratio of the number of missed detections to the total number of transmitters.

## 3. DSLoc Methodology

This section introduces the proposed DSLoc model’s network architecture and processing steps. As shown in Figure 1, DSLoc consists of the following parts: the feature enhancement module, the HRNet network, and the heatmap centroid localization. We will introduce each part separately.

### 3.1. Feature Enhancement Module

The purpose of this module is to enhance the features of the initial input sensor matrix. Due to various factors such as different sensor deployment positions (e.g., dynamic movement, rooftops, building sides), differences introduced by heterogeneous sensors, and occasional RSS anomalies, the raw measured RSS is often not accurate enough. We aim to perform some bias correction on the RSS measurements from sensors at different locations. After correction, the sensor matrix remains extremely sparse. For instance, if 15 sensors are deployed within a 384 × 384 grid, approximately only 0.01% of the pixel values are non-zero. The sparse input matrix makes it very challenging for the neural network to learn effectively. The main tasks of this module are RSS correction and handling the sparsity.

#### 3.1.1. Bias Correction

We use a pixel attention-based [24] scaling method for RSS matrix bias correction. This approach, which uses a pixel attention convolutional neural network, allows for more detailed adjustments of the RSS matrix to change the attention to different locations, thereby enhancing localization accuracy. As shown in Figure 2, a two-channel convolutional neural network is used for feature extraction. Channel 1 is represented as follows:(3)$$X_{1}=f^{13\times 13}(\delta (f^{13\times 13}(x))),$$
Channel 2 is represented as follows:(4)$$X_{2}=f_{2}^{31\times 31}(\delta (f^{31\times 31}(x))),$$
where x is the input matrix, $X_{1}$ and $X_{2}$ are the output matrix of the corresponding channel, $f^{13\times 13}(\cdot )$ denotes convolution operation with a 13 × 13 kernel, δ represents the ReLU function, and $f_{2}^{31\times 31}(\cdot )$ denotes a dilated convolution [25] operation with a 31 × 31 kernel [26] and a dilation rate of 2. Channel 1 extracts local features of the matrix, while Channel 2 uses a large convolutional kernel and employs dilated convolution in the second layer to extract sparse features from larger feature layers. Then, a 1 × 1 convolution is used to merge the two channels into one, and the Sigmoid function adjusts the output values between 0 and 1. This scales the original RSS matrix, achieving sensor RSS correction without altering the original sensor matrix distribution.

#### 3.1.2. Sparse Expansion

To effectively handle sparse RSS matrices, we have introduced a novel approach inspired by heatmap analysis techniques. In contrast to previous direct mapping methods, we have, for the first time, applied Gaussian convolution kernels in the preprocessing stage of the RSS matrix to enhance data continuity and interpretability, as shown in Figure 3. A 2D Gaussian distribution convolution kernel is used. In this example, a 128 × 128 grid matrix is convolved with a 7 × 7 kernel with a variance of 1. This process is performed offline and does not require gradient propagation. As the grid number increases, we recommend using larger convolution kernels with increased variance. For example, in a 384 × 384 grid, a 23 × 23 convolution kernel with a variance of 3 is effective.

### 3.2. HRNet

The HRNet is a network structure designed to address the issue of resolution loss in traditional convolutional neural networks, commonly applied in tasks such as human pose estimation [27] and semantic segmentation. The network structure, as shown in Figure 4, offers advantages over other network models as it maintains high-resolution feature maps throughout, preserving more detailed information from the input matrix. The HRNet employs a parallel branch design to fuse features from different scales, capturing both local and global features. By maintaining high-resolution feature maps and utilizing a parallel branch design, HRNet ensures that detailed information from the input is preserved and effectively used. This enhances the model’s ability to accurately localize transmitters even in sparse sensor environments, making it particularly well-suited for our application.

The specific structure of HRNet used in our approach is as follows: features are extracted using 5 × 5 convolutions. To reduce model complexity, we employ depthwise separable convolutions instead of the original convolutional structure. In the feature extraction stage of the model input, two downsampling processes are initially applied, with 1/4 input size into the network shown in Figure 4. Multiple feature maps of different sizes are generated at the output of the network. These are fused using the approach depicted in Figure 5, followed by a 1 × 1 convolution to transform them into a single-channel representation. Subsequently, a four-fold upsampling is performed to obtain a heatmap predicting the transmitter’s location, matching the input size.

### 3.3. Heatmap Centroid Localization

Heatmap regression [23] is widely used in tasks such as human pose estimation, where keypoints are identified by locating peaks using argmax approach. While effective for small-scale data, this method suffers from significant localization errors in large-scale transmitter localization scenarios. Upon analysis, as shown in the Figure 6, peaks in the heatmap do not always coincide with the predicted centers of Gaussian distributions. This discrepancy arises because existing models cannot perfectly fit the Gaussian distribution within the heatmap, especially when the variance of the Gaussian distribution is large.

Although the predicted Gaussian distributions in the heatmap may not be perfectly accurate, their overall distribution characteristics are quite close. Therefore, we use centroid computation to determine keypoint locations, reducing occasional errors inherent in the argmax approach. The formula for computing the image centroid is as follows:(5)$$(x_{c},y_{c})=\frac{\left(\sum_{j=1}^{n}\sum_{i=1}^{n}i*P_{ij},\sum_{j=1}^{n}\sum_{i=1}^{n}j*P_{ij}\right)}{\sum_{j=1}^{n}\sum_{i=1}^{n}P_{ij}},$$
where $P_{ij}$ represents the pixel value at position ( i,j) in the heatmap, and *i*, *j* are the horizontal and vertical coordinates, respectively. In our experiments, *n* denotes the number of divisions for both the width and height of the grid, which are equal. As illustrated in Figure 6, keypoints determined by centroid computation are closer to the true positions in the ground truth. The centroid localization method represents an improvement over the argmax approach [28], which locates keypoint locations to find peaks. Within the centroid localization framework, the localization error is measured by the Euclidean distance between the centroids of the target heatmap and the predicted heatmap. The formula is as follows:(6)$$d=\sqrt{(x_{t}-x_{p}{)}^{2}+(y_{t}-y_{p}{)}^{2}},$$

Here, $\left(x_{t},y_{t}\right)$ denotes the coordinates of the point in the ground truth, and $\left(x_{p},y_{p}\right)$ denotes the coordinates of the predicted point.

## 4. Experimental Design and Results

In this section, we designed experiments to evaluate our proposed DSLoc model, comparing its performance in miss rate and mean localization error against the SOTA method, while exploring other aspects of DSLoc’s capabilities.

### 4.1. Experimental Setup and Dataset

The experiments were conducted using a model implemented in PyTorch, optimized for accelerated processing on Graphics Processing Units (GPUs). The hardware setup consisted of a single CPU Intel Core i9-12900k and an NVIDIA GeForce RTX4090 GPU manufactured in Taiwan. The model was trained for 1000 epochs with a batch size of 64, using the Adam optimizer with a learning rate of 0.0002 and MSE for loss function.

We utilized a dataset collected from large-scale outdoor environments using the POWDER Testbed at the University of Utah [29]. The data consist of RSS measurements taken using a variety of sensors in a 2 km × 2 km area, with 10–25 sensors per measurement. Each sample includes RSS values from both stationary and mobile receivers. Most of the stationary sensors were fixed at endpoints such as building sides, rooftops, towers, or standalone poles, while a smaller subset of sensors was mounted on shuttles that traveled specific routes throughout the campus. The dataset comprises 5214 unique samples with transmitters, each with unique transmitter positions. The majority of samples contain a single transmitter, with a small subset including two transmitters (specifically 346 samples, only used exclusively in extension experiments). The receivers used in this dataset include Ettus USRP X310 and B210 radios manufactured by Ettus Research in California, USA. Due to equipment variations, all RSS values in the dataset are uncalibrated. The environmental conditions during data collection varied, encompassing different times of day and various weather conditions, thereby providing a comprehensive dataset for testing localization methods under realistic and diverse scenarios.

To balance between localization accuracy and computational efficiency, we adjusted the size of the input grid matrix. Larger grid sizes provide finer pixel resolution in the actual coverage area. However, with a fixed number of sensors, the density of the sensor grid decreases accordingly, posing challenges for model training. The network was trained to regress from the input RSS matrix to the heatmap. The dataset is displayed as input, output in Figure 1 above. The dataset was split into training and testing sets in an 8:2 ratio. To avoid accidental errors, we randomly split the dataset ten times in the experiment, and comprehensively considered all divided test sets for verification.

### 4.2. Performance Comparisons

As shown in Figure 7a, compared to DeepTxFinder [19], DeepMTL [20], and TL;DL [21], DSLoc consistently exhibits lower mean localization errors across different grid divisions. As the number of grid divisions increases, the pixel density occupied by sensors decreases, leading to significantly larger localization errors in other models. However, DSLoc is less affected by grid division changes, maintaining a relatively stable mean localization error. Figure 7b illustrates that, except for DeepTxFinder, which has no miss rate due to its fully connected layer outputting the transmitter position, the miss rate trends of the other models closely follow the trend of the localization error curves in Figure 7a.

We comprehensively considered both miss rate and mean localization error. Given the limited number of our dataset, using a larger number of grid divisions increases the likelihood of encountering out-of-distribution (OOD) scenarios, leading to a higher miss rate and consequently larger mean localization errors. The DSLoc model reduces the miss rate, resulting in a lower mean localization error. Theoretically, we aim to use a higher number of grid divisions for finer actual area mapping per pixel, which should reduce the mean localization error. However, this requires a sufficiently large training dataset. In our experiments, DSLoc alleviated the miss rate caused by insufficient training data but did not completely eliminate it. Using a large number of grid divisions remains meaningful, as it provides more accurate localization when the transmitter is detected. Therefore, the mean localization error curve for DSLoc is smoother than the miss rate curve. The visualization analysis, as depicted in Figure 8, showcases two scenarios. Figure 8a illustrates the optimal performance of the model, where the predicted heatmap closely aligns with the ground truth, indicating high accuracy in localization. Conversely, Figure 8b highlights instances of localization error, where there is a discrepancy in distance between the predicted transmitter and the true transmitter location, or where the probabilities in the predicted heatmap are relatively low, both of which contribute to error margins in the localization process.

We categorized the potential sources of faults in the localization process and assessed their impact on overall performance. The primary sources of errors identified are as follows:Grid Division Granularity: Finer grid divisions result in lower pixel densities occupied by sensors, potentially leading to increased localization errors due to insufficient resolution in other models. It was observed that DSLoc maintains stable performance across a range of grid divisions, effectively mitigating this issue.Dataset Limitations: The limited size of the dataset impacts the model’s generalization ability. With a smaller dataset, there is a greater probability of encountering out-of-distribution scenarios, resulting in higher miss rates.Hardware Variations: Disparities in receiver gain settings and the absence of calibration data introduce additional uncertainties into RSS measurements. Uncalibrated values affect the relative comparisons between different receivers, contributing to localization error. Although DSLoc compensates for such variations to a certain degree, hardware-related inconsistencies remain a significant source of error.

Furthermore, the density of WSNs and environmental factors were found to influence positioning performance. Notably, DSLoc exhibits substantial enhancements in terms of mean localization error and miss rate when compared to alternative models.

### 4.3. Ablation Experiment

#### 4.3.1. Evaluation of Feature Enhancement Module

To assess the impact of the feature enhancement module on transmitter localization, we compared the DSLoc model with another network model identical in structure but without the feature enhancement module. As shown in Figure 9, the curves labeled “Without Feature Enhancement Module” showed higher localization errors and miss rates compared to the DSLoc model, especially in scenarios with a large number of grid divisions. The experiments demonstrate that the feature enhancement module improves localization accuracy and offers better adaptability to sparse pixel matrices in high grid division scenarios. 

#### 4.3.2. Comparison of HRNet and U-Net

Previous SOTA localization models [20,21] often employed the U-net architecture [30]. In this paper, we introduce the HRNet structure, which is used for human skeleton keypoint detection, into transmitter localization. To verify its effectiveness, we compare DSLoc using HRNet with a model where HRNet is replaced by U-net. As shown in Figure 8, HRNet reduces both mean localization error and miss rate across all configurations compared to the U-net network. The experimental results demonstrate that HRNet performs better in heatmap regression for transmitter localization.

#### 4.3.3. Evaluation of Centroid Localization

Locating the transmitter position in the heatmap using argmax and centroid methods does not require gradient propagation, so it does not affect the miss rate of the trained network. The difference lies only in the localization error. As shown in Figure 10, the centroid-based localization method achieves a smaller mean localization error compared to the argmax method, with an improvement ratio of approximately 6%–12%.

### 4.4. Extension Experiment

To fully explore the performance of DSLoc in transmitter localization, we designed a dual-transmitter localization experiment. Like the multiple output channel heatmaps in human skeleton keypoint detection, dual transmitters correspond to two output channels. We measured the mean localization error and miss rate for different grid divisions. As shown in Table 1, DSLoc is effective for dual transmitters but exhibits higher mean localization error and miss rate compared to single transmitter scenarios. Our analysis found that the limited number of dual-transmitter samples hinders the model’s ability to generalize and accurately localize in the current scenarios. Therefore, expanding the dataset to include a greater volume of dual-transmitter samples is significant for reducing localization errors and unlocking the full potential of DSLoc. The DSLoc provides a new approach for deep learning in multi-transmitter localization.

## 5. Conclusions and Future Work

This paper presents DSLoc, a novel deep learning-based approach for transmitter localization in sparse WSNs. By leveraging the improved HRNet and designing efficient feature enhancement modules, DSLoc addresses the challenges posed by sparse sensor deployments over large areas. Utilizing a heatmap-based approach and calculating the centroid of the predicted regions, DSLoc achieves precise transmitter localization even in low-density WSNs. The extensive experiments conducted on WSNs with varying densities, including comparison, ablation, and expansion experiments, demonstrate DSLoc’s superior performance in terms of both localization accuracy and scalability.

Despite these significant achievements, there are the following areas for future work to further enhance localization capabilities:Multi-Transmitter Scenarios: While DSLoc performs well with single transmitter scenarios, future research could focus on optimizing the model to handle multiple transmitters simultaneously. This research includes improving the model’s ability to distinguish and accurately localize multiple signal sources in close proximity.Dataset Expansion: Expanding the dataset to include a wider variety of sensor types and more complex environments will improve the model’s generalization ability. Incorporating data from urban, suburban, and rural areas, as well as different climatic conditions, will make the model more robust and versatile.Integration with Other Sensing Modalities: Combining other sensing technologies, such as drones or mobile sensors, could provide a more comprehensive and dynamic solution for transmitter localization. This multi-modal approach could leverage the strengths of different technologies to achieve even higher localization accuracy and coverage.

In conclusion, DSLoc represents a significant advancement in the field of transmitter localization in sparse WSNs, particularly HRNet, efficient feature enhancement module, and centroid-based localization method usage, which highlights its potential to be adapted and integrated into existing WSN frameworks. This adaptability can lead to significant cost savings and operational efficiencies. Its innovative design and superior performance make it a promising solution for various applications requiring accurate and robust signal source tracking. Future work will focus on addressing the outlined challenges, further enhancing location capabilities, and broadening its applicability in real-world scenarios.

## Figures and Tables

**Figure 1 sensors-24-05335-f001:**
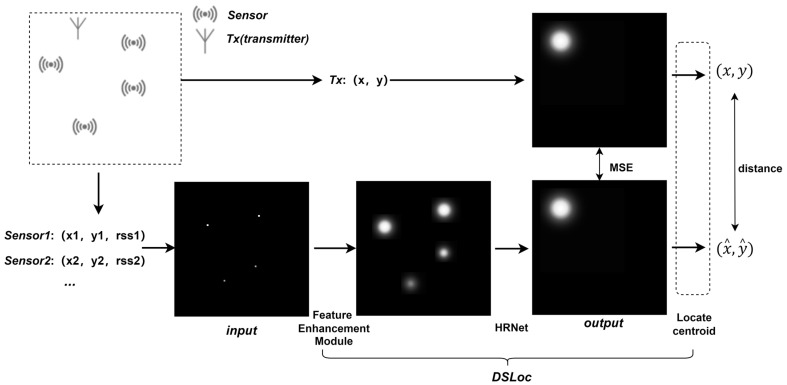
Transmitter localization flowchart (number of sensors and Gaussian size used for demonstration purposes only).

**Figure 2 sensors-24-05335-f002:**
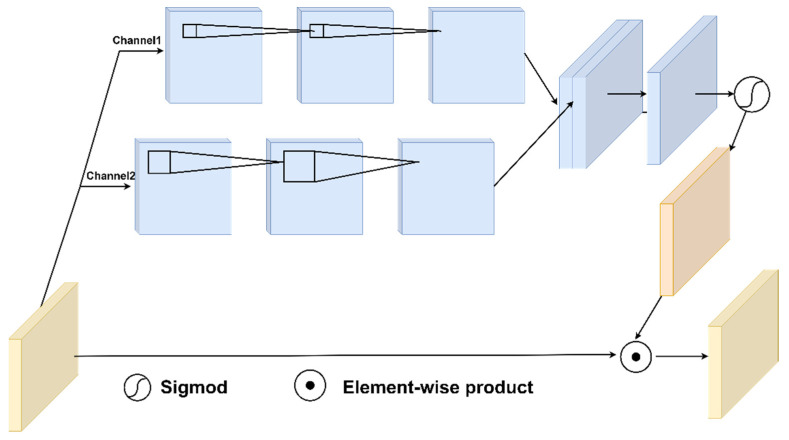
Pixel attention module for bias correction.

**Figure 3 sensors-24-05335-f003:**
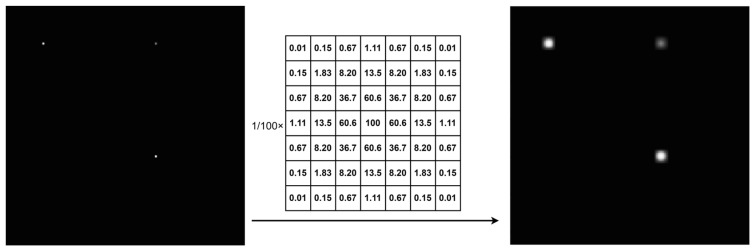
Gaussian convolution process (number of sensors used for demonstration purposes only).

**Figure 4 sensors-24-05335-f004:**
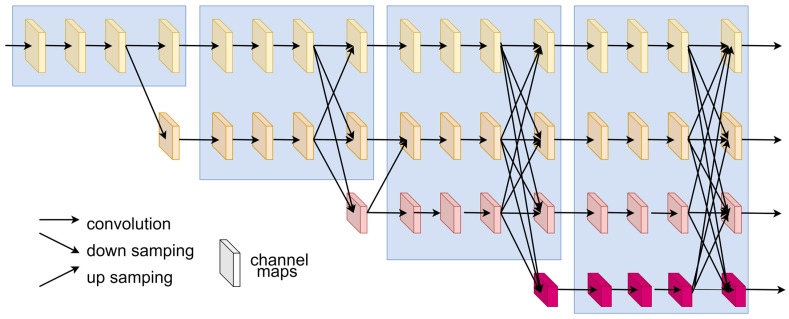
HRNet architecture (the squares of different colors represent channel maps of different sizes. The whole system consists of four stages, and each stage performs a feature extraction transformation).

**Figure 5 sensors-24-05335-f005:**
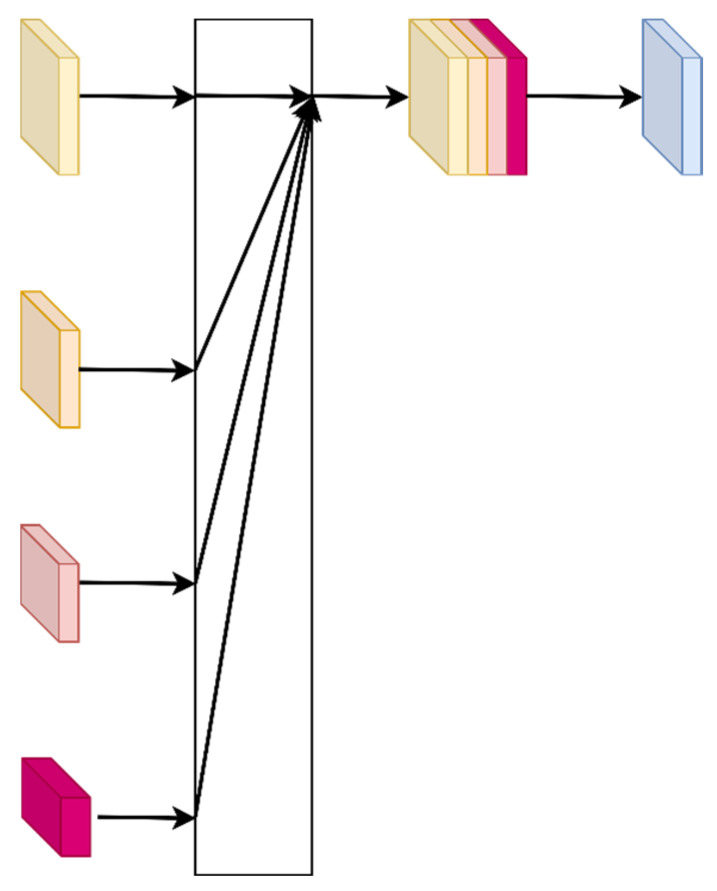
Multi-Scale Output Fusion Module (the components have been defined in Figure 4, and the feature maps of different sizes are fused by first upsampling separately and then feature concatenation and fusion).

**Figure 6 sensors-24-05335-f006:**
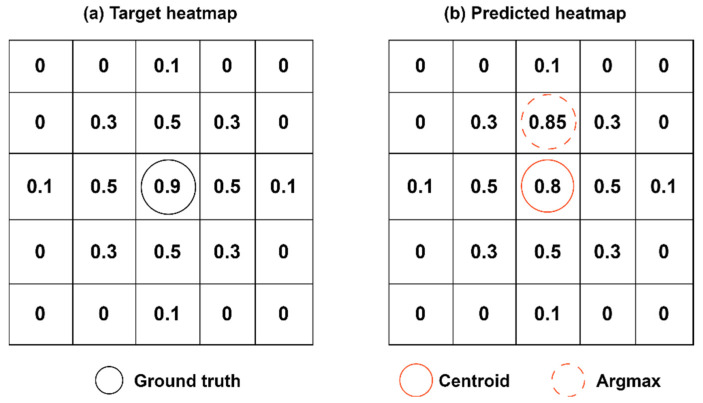
Centroid vs. Argmax approach.

**Figure 7 sensors-24-05335-f007:**
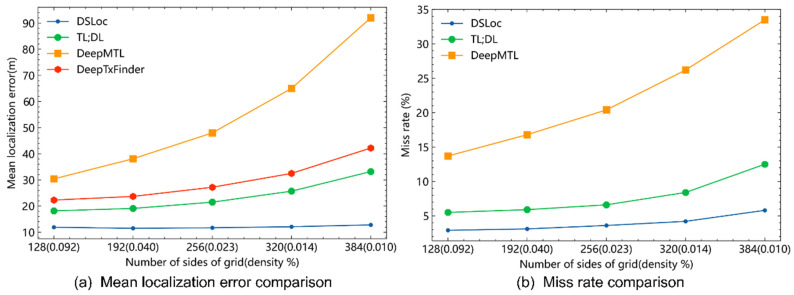
Performance comparison of different algorithms.

**Figure 8 sensors-24-05335-f008:**
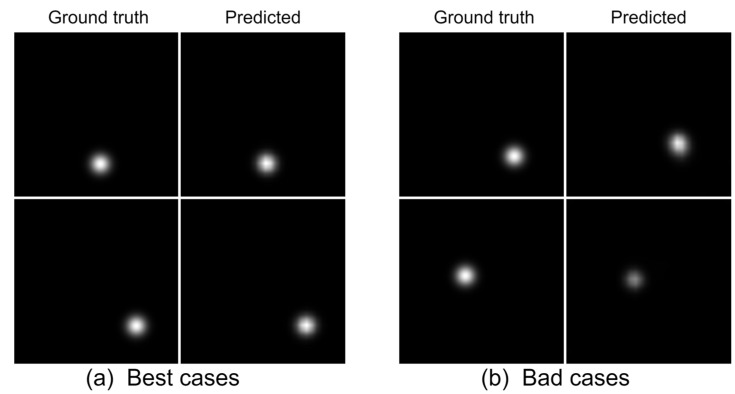
Visualization of localization cases.

**Figure 9 sensors-24-05335-f009:**
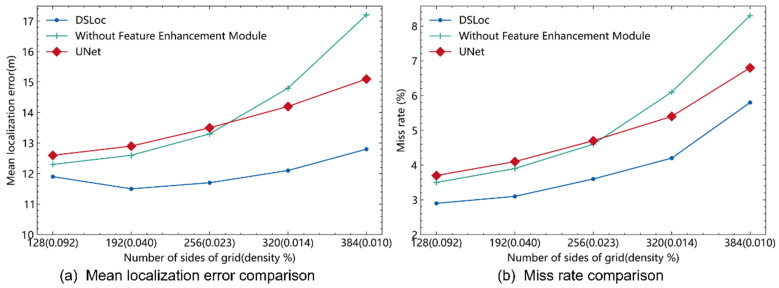
Performance comparison of ablation experiment.

**Figure 10 sensors-24-05335-f010:**
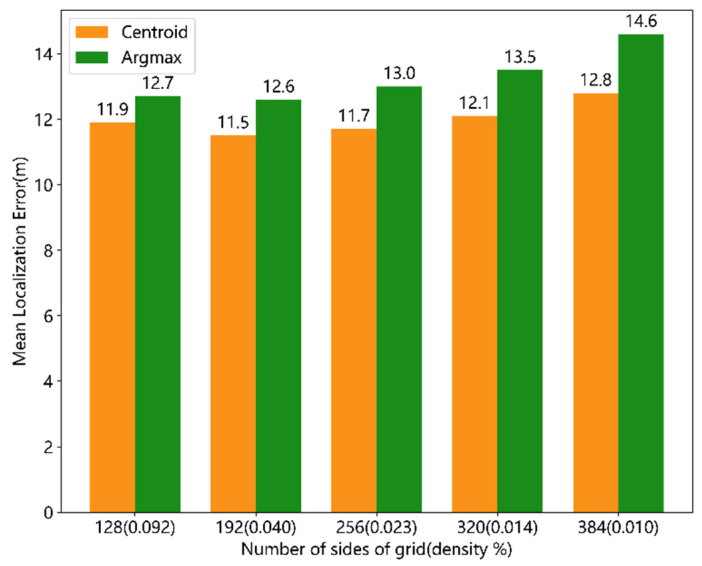
Performance comparison of Centroid and Argmax.

**Table 1 sensors-24-05335-t001:** Dual transmitter experimental results.

Number of Divided Grids	Mean Localization Error (m)	Missing Rate (%)
128	23.4	7.3
256	32.8	10.7
384	47.4	15.2

## Data Availability

The original data presented in the study are openly available in the dataset collected from large-scale outdoor environment at reference [29].

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
