# Peer review of "Deep Learning-Based Transmitter Localization in Sparse Wireless Sensor Networks"

_sensors, 2024, doi:10.3390/s24165335_

Round 1

Reviewer 1 Report

Comments and Suggestions for Authors

After reviewing the article “Deep Learning-Based transmitter localization in sparse wire-2 less sensor networks” I have the following comments:

1.      In the introduction section, the author could provide a more concise and explicit explanation of the reason and importance of the proposed DSLoc approach. At present, there is a lack of a concise explanation about the research gap and how our work specifically solves that gap distinctively when compared to existing methodologies.

2.      Similarly, the related works section should present a comprehensive evaluation of existing approaches, particularly focusing on their weaknesses and how DSLoc addresses and surpasses them. Providing a more comprehensive analysis in comparison to the most advanced approaches in the literature would enhance the quality of the study. The authors can also refer to “Extended Kalman Filter-based Localization Algorithm by Edge Computing in Wireless Sensor Networks”, “Optimizing Network Lifespan through Energy Harvesting in Low-Power Lossy Wireless Networks,” and so on.

3.      The paper provides a quick overview of the feature improvement module and the HRNet architecture, but it does not offer a thorough explanation for the selection of these particular methods over alternative approaches.

4.      The experimental setup section should provide comprehensive details regarding the dataset, such as the specific types of sensors employed, their distribution, and the prevailing ambient conditions.

5.      The article lacks a sensitivity analysis of the crucial parameters employed in the model, such as grid size, learning rate, and convolution kernel size. An investigation of this nature would be essential for comprehending the resilience and dependability of the DSLoc approach across various setups.

6.      Although the paper includes a comparison of DSLoc with other approaches, it does not provide a thorough statistical analysis of the data. By using metrics such as standard deviation and performing significance tests, the comparisons would be enhanced in terms of rigor and credibility.

7.      The authors must provide a comprehensive error analysis section elucidating the possible origins of faults in the localization process and their impact on the overall performance. This may involve a detailed analysis of errors categorized by certain scenarios or types of environments.

8.      The authors should consider additional visualizations, such as heatmaps comparing projected and actual transmitter locations and graphical representations of the distribution of localization errors, which would improve the understanding and significance of the experimental findings.

9.      The authors should revise the article carefully for grammatical mistakes and typo errors.

10.   The existing references seem very limited, more up-to-date references should be added.

Comments on the Quality of English Language

Extensive editing of the English language required

Author Response

Comments 1: In the introduction section, the author could provide a more concise and explicit explanation of the reason and importance of the proposed DSLoc approach. At present, there is a lack of a concise explanation about the research gap and how our work specifically solves that gap distinctively when compared to existing methodologies.
Response 1: Thank you for pointing this out. We agree with this comment. Therefore, we have thoroughly revised and expanded upon the introduction in lines 76-94 of the manuscript. In this updated section, we now explicitly delineate the existing research gap in the domain of deep learning-based transmitter localization. We emphasize the shortcomings of current methodologies and outline how our DSLoc framework uniquely addresses these limitations, thereby contributing to a more robust and accurate localization solution. We have also included a brief comparative analysis against prevalent techniques to underscore the distinct advantages and innovations of our approach. By doing so, we aim to provide readers with a clearer understanding of the context and novelty of our work.

Comments 2: Similarly, the related works section should present a comprehensive evaluation of existing approaches, particularly focusing on their weaknesses and how DSLoc addresses and surpasses them. Providing a more comprehensive analysis in comparison to the most advanced approaches in the literature would enhance the quality of the study. The authors can also refer to “Extended Kalman Filter-based Localization Algorithm by Edge Computing in Wireless Sensor Networks”, “Optimizing Network Lifespan through Energy Harvesting in Low-Power Lossy Wireless Networks,” and so on.
Response 2: Thank you for pointing this out. We agree with this comment. Therefore, we have revised and improved lines 40-46 and 76-94 in the manuscript. We fully understand that the references you pointed out represent an alternative research path, which provides a beneficial contrast to our approach of using deep learning for transmitter localization. In the revision of our paper, we have incorporated these references into the discussion section to provide a more comprehensive overview of the research background in this field. While we agree that comparing different methods can enhance understanding of the topic, since this approach is a completely different line of technology from this paper, we do not directly compare with the studies mentioned in our experimental analysis at this stage. This limitation has been acknowledged in the manuscript, along with our intention to consider this aspect in future research. We hope you will understand this situation and look forward to further guidance from you.

Comments 3: The paper provides a quick overview of the feature improvement module and the HRNet architecture, but it does not offer a thorough explanation for the selection of these particular methods over alternative approaches.
Response 3: Thank you for pointing this out. We agree with this comment. In order to more adequately explain the reasoning behind our selection of the feature enhancement module and the HRNet structure, we have made corresponding revisions and additions in the manuscript. Specifically, we have elaborated on the design concept of the feature enhancement module between lines 133 to 137 and lines 150 to 155. From lines 162 to 172, we have explicated the theoretical foundations of the HRNet structure as well as its advantages in practical applications. We hope that these amendments not only clarify the rationale for choosing specific architectures but also better demonstrate our research contributions.

Comments 4: The experimental setup section should provide comprehensive details regarding the dataset, such as the specific types of sensors employed, their distribution, and the prevailing ambient conditions.
Response 4: Thank you for pointing this out. We agree with this comment. Therefore, we have revised and improved the comprehensive and detailed information about the dataset in lines 223-236 of the manuscript.

Comments 5: The article lacks a sensitivity analysis of the crucial parameters employed in the model, such as grid size, learning rate, and convolution kernel size. An investigation of this nature would be essential for comprehending the resilience and dependability of the DSLoc approach across various setups.
Response 5: Thank you for pointing this out. We agree with this comment. Therefore, we have improved the analysis part again. Figures 7 and 9 in the manuscript have contained the analysis of the influence of grid size on the use of the model. The influence of convolution kernel size is discussed in lines 153-158, and the learning rate has no specific influence on the results to some extent.

Comments 6: Although the paper includes a comparison of DSLoc with other approaches, it does not provide a thorough statistical analysis of the data. By using metrics such as standard deviation and performing significance tests, the comparisons would be enhanced in terms of rigor and credibility.
Response 6: Thank you for pointing this out. We agree with this comment. Therefore, we need to explain the problem you raised. We chose the mean localization error, and the mean localization error and standard deviation have the same function here. In the past work, most of them used the mean localization error, and we followed this method. We additionally chose the missed rate as our evaluation index and made an in-depth analysis, and at the same time increase the rigor and reliability of the model, specifically in lines 256-259 and 266-273 of the manuscript.

Comments 7: The authors must provide a comprehensive error analysis section elucidating the possible origins of faults in the localization process and their impact on the overall performance. This may involve a detailed analysis of errors categorized by certain scenarios or types of environments.
Response 7: Thank you for pointing this out. We agree with this comment. Therefore, in the revised manuscript, we improved the source and influence of the error in the localization process and made a detailed analysis, specifically in lines 274-291.

Comments 8: The authors should consider additional visualizations, such as heatmaps comparing projected and actual transmitter locations and graphical representations of the distribution of localization errors, which would improve the understanding and significance of the experimental findings.
Response 8: Thank you for pointing this out. We agree with this comment. Therefore, we have increased the localization visualization and concrete analysis, and improved the understanding and significance of the experimental results, specifically in lines 266 -273 of the manuscript and Figure 8.

Comments 9: The authors should revise the article carefully for grammatical mistakes and typo errors.
Response 9: Thank you for pointing this out. We agree with this comment. Therefore, we corrected grammatical mistakes and typo errors in the revised manuscript.

Comments 10: The existing references seem very limited, more up-to-date references should be added.
Response 10: Thank you for pointing this out. We agree with this comment. Therefore, we added and updated references, 3, 4, 9 and 15-17 to provide a more comprehensive overview of the research background in this field.

Reviewer 2 Report

Comments and Suggestions for Authors

This paper proposed deep Learning-based transmitter localization in sparse wireless sensor networks. The paper is well-written and organized. The paper's contribution, if any, has not been significant. However, the paper needs major revision before being accepted for publication in Sensors, as follows:

1.      In the abstract, change the sentence "To address sparse data,….."  to "To address localization in sparse wireless sensor networks,….. "

2.      The type of deep learning should be clarified in the abstract and the introduction section.

3.      Add significant results to the abstract in terms of localization error.

4.      The introduction section could discuss several references using AI-based localization algorithms to provide a comprehensive overview.

5.      Highlight the contribution of the paper at the end of the introduction section.

6.      Define the abbreviation "DSLOC" for the first appearance.

7.      The hyperparameters and flowcharts of the adopted deep learning must be presented.

8.      The parts inside Figures 4 and 5 should be indicated.

9.      Did the authors implement the deep learning models (i.e., DeepTxFinder,  DeepMTL, TL, and DL) in section 4.2? Please clarify that.

10.   The mean localization error in Figures 7, 8, 9, and Table 1 should be clarified. It is estimated for x or y or average x and y.

11.   Is the proposed method considered efficient given that the error rate is significant, exceeding 10 meters, as presented in Figure 8a, Figure 9, and Table 1? It is important to note that this error level is unacceptable in certain applications, such as patient localization or target tracking.

12.   Although the authors suggested important future directions in the conclusion section, the conclusion is somewhat brief. It could benefit from a more detailed discussion of the impact of the findings.

13.   Some references are outdated. It should be updated (e.g., ref. 3).

Comments on the Quality of English Language

Minor editing of English language required

Author Response

Comments 1: In the abstract, change the sentence "To address sparse data,….."  to "To address localization in sparse wireless sensor networks,….. ".
Response 1: Thank you for pointing this out. We agree with this comment. Therefore, we have revised the expression you mentioned in the abstract.

Comments 2: The type of deep learning should be clarified in the abstract and the introduction section.
Response 2: Thank you for pointing this out. We agree with this comment. Therefore, in the abstract and introduction, we refine the deep learning methods used, specifically in lines 15-16 and 81-82 of the manuscript.

Comments 3: Add significant results to the abstract in terms of localization error.
Response 3: Thank you for pointing this out. We agree with this comment. Therefore, in the abstract, we improve the expression of the model's promotion of localization error, specifically in lines 18-22 of the manuscript.

Comments 4: The introduction section could discuss several references using AI-based localization algorithms to provide a comprehensive overview.
Response 4: Thank you for pointing this out. We agree with this comment. We need to explain your suggestion. Lines 47 -75 of our manuscript are an overview of localization methods based on artificial intelligence. In the revised manuscript, we have further improved the description of the limitations of existing methods based on artificial intelligence and the improvement of our proposed methods.

Comments 5: Highlight the contribution of the paper at the end of the introduction section.
Response 5: Thank you for pointing this out. We agree with this comment. Therefore, the introduction part of the article is revised to highlight the contribution of the paper, which is in the manuscript from 81 lines to 94 lines.

Comments 6: Define the abbreviation "DSLoc" for the first appearance.
Response 6: Thank you for pointing this out. We agree with this comment. Therefore, the first use of DSLoc was modified, which is on lines 14-15 and 81-82 of the manuscript. DSLoc is the meaning of "Deep Learning-Based in sparse wireless sensor networks transmitter localization method", and it is the name of the method in this paper, not as an abbreviation.

Comments 7: The hyperparameters and flowcharts of the adopted deep learning must be presented.
Response 7: Thank you for pointing this out. We agree with this comment. Therefore, the flowcharts of the deep learning used in the model are shown in Figure 1, Figure 4 and Figure 5, and the hyperparameters used in the model are defined in lines 218 -222 of the manuscript.

Comments 8: The parts inside Figures 4 and 5 should be indicated.
Response 8: Thank you for pointing this out. We agree with this comment. Therefore, in the revised manuscript, the descriptions of components used in Figures 4 and 5 have been improved. We have also added the description of Figure 4 and Figure 5, specifically in lines 183 -185 and 187 -189 of the manuscript.

Comments 9: Did the authors implement the deep learning models (i.e., DeepTxFinder,  DeepMTL, TL, and DL) in section 4.2? Please clarify that.
Response 9: Thank you for pointing this out. We agree with this comment. Therefore, the contrast model used in the experimental part has been defined in lines 245 -246.

Comments 10: The mean localization error in Figures 7, 8, 9, and Table 1 should be clarified. It is estimated for x or y or average x and y.
Response 10: Thank you for pointing this out. We agree with this comment. The mean localization error used in this paper is defined in lines 108-112, and the mean localization error is the Euclidean distance between two points as shown in formula (5) in the manuscript.

Comments 11: Is the proposed method considered efficient given that the error rate is significant, exceeding 10 meters, as presented in Figure 8a, Figure 9, and Table 1? It is important to note that this error level is unacceptable in certain applications, such as patient localization or target tracking.
Response 11: Thank you for pointing this out. We need to explain your question here. The scene this paper faces is spectrum monitoring, locating illegal transmitters. In an area of 2Km*2Km, such as a school, the transmitter can be located with a precision of 10m, which has met the needs of the scene. The density of WSNs used in this paper is extremely small. If an increased sensor is deployed or enough data is collected, it can achieve higher accuracy, which can be used for reference in patient positioning and other scenes.

Comments 12: Although the authors suggested important future directions in the conclusion section, the conclusion is somewhat brief. It could benefit from a more detailed discussion of the impact of the findings.
Response 12: Thank you for pointing this out. We agree with this comment. Therefore, In the revised manuscript, we have perfected the conclusion part, which is in lines 359 -367.

Comments 13: Some references are outdated. It should be updated (e.g., ref. 3).
Response 13: Thank you for pointing this out. We agree with this comment. Therefore, we added and updated references, 3, 4, 9 and 15-17, and made some more detailed analysis to provide a more comprehensive overview of the research background in this field, in lines 43 -46.

Round 2

Reviewer 1 Report

Comments and Suggestions for Authors

No further comments from my side. The article looks good in its current form. 

Comments on the Quality of English Language

Minor editing of English language required

Author Response

Thank you very much for your review of the article. We revised it again according to the advice of the academic editor.

Reviewer 2 Report

Comments and Suggestions for Authors

The authors addressed all my previous comments. Therefore,  the paper can be accepted for publication in the current form. 

Comments on the Quality of English Language

None

Author Response

(The authors gave the same response as above.)
